# The Diabetic Cardiac Fibroblast: Mechanisms Underlying Phenotype and Function

**DOI:** 10.3390/ijms21030970

**Published:** 2020-02-01

**Authors:** Scott P. Levick, Alexander Widiapradja

**Affiliations:** 1Kolling Institute for Medical Research, Royal North Shore Hospital, St Leonards 2065, Australia; Alexander.widiapradja@sydney.edu.au; 2Faculty of Medicine and Health, The University of Sydney, St Leonards 2065, Australia

**Keywords:** diabetic cardiomyopathy, heart, diabetes, fibrosis, high glucose

## Abstract

Diabetic cardiomyopathy involves remodeling of the heart in response to diabetes that includes microvascular damage, cardiomyocyte hypertrophy, and cardiac fibrosis. Cardiac fibrosis is a major contributor to diastolic dysfunction that can ultimately result in heart failure with preserved ejection fraction. Cardiac fibroblasts are the final effector cell in the process of cardiac fibrosis. This review article aims to describe the cardiac fibroblast phenotype in response to high-glucose conditions that mimic the diabetic state, as well as to explain the pathways underlying this phenotype. As such, this review focuses on studies conducted on isolated cardiac fibroblasts. We also describe molecules that appear to oppose the pro-fibrotic actions of high glucose on cardiac fibroblasts. This represents a major gap in knowledge in the field that needs to be addressed.

## 1. Introduction

Diabetic individuals suffer from multiple cardiovascular complications including coronary artery disease, hypertension, and renal disease. As such, individuals with diabetes have a 20%–40% higher incidence of heart failure [1] and a 33% higher risk of hospitalization [2]. Separate from these aforementioned cardiovascular comorbidities, there exists a specific diabetic cardiomyopathy characterized in part by cardiomyocyte hypertrophy and interstitial cardiac fibrosis [3,4,5]. This concept was originally identified by Rubler et al. [6] in 1972, after case study observations of cardiomegaly, cardiac fibrosis, small vessel alterations, and heart failure in diabetic individuals. While each of these structural abnormalities contribute to the eventual development of heart failure, cardiac fibrosis, which is characterized by an increase in extracellular matrix (ECM) proteins, particularly collagen types I and III, leads to impaired left-ventricular compliance and diastolic dysfunction, ultimately manifesting as heart failure with preserved ejection fraction (HFpEF) [7,8,9,10,11,12]. Despite the increased morbidity and mortality associated with diabetes-induced cardiac fibrosis [13], there are no effective treatment strategies to ameliorate or reverse this fibrosis and subsequent HFpEF.

One potential cellular target for anti-fibrotic therapies for the diabetic heart is the cardiac fibroblast, the final effector cell in the fibrotic process, responsible for generating and maintaining the ECM. Maintenance of the ECM by these cells through a balance in ECM synthesis and degradation contributes significantly to the normal structure and function of the myocardium, providing alignment of cardiomyocytes, prevention of cardiomyocyte slippage, transduction of force, protection from cardiomyocyte overstretching, and tensile strength [14,15]. However, under pathological conditions, adverse alterations to the mechanical properties and the biochemical environment of the heart can result in the activation of cardiac fibroblasts and a pro-fibrotic phenotype [8,9,10,16].

This review article is focused on the cardiac fibroblast and the mechanisms that govern the promotion of a pro-fibrotic phenotype of these cells in response to the “diabetic” environment. Due to the limited ability to specifically investigate cardiac fibroblasts in vivo, most of the experiments discussed in this review are isolated cell cultures of normal cardiac fibroblasts from various species, then exposed to high-glucose (HG) conditions, as well as cardiac fibroblasts isolated from rodent models of diabetes.

## 2. Cardiac Fibroblast Phenotype

### 2.1. Extracellular Matrix Production

The ECM components primarily responsible for functional changes in the heart are the fibrillar collagens, specifically collagen I and III. HG conditions induce excess collagen synthesis by isolated neonatal and adult rat, neonatal and adult mouse, and human cardiac fibroblasts, which is not due to the effect of HG to increase osmolarity [17,18,19,20,21,22,23,24,25]. This increased collagen production is also true of cardiac fibroblasts isolated from an in vivo diabetic state and then cultured. For example, when compared to fibroblasts from non-diabetic individuals, collagen I production was increased from isolated right-atrial cardiac fibroblasts obtained from diabetic individuals without left-ventricular dysfunction, undergoing coronary artery bypass graft surgery [26]. Similarly, cardiac fibroblasts isolated from diabetic rodent models produce excess ECM when cultured. For example, cardiac fibroblasts isolated from Zucker obese diabetic rats showed increased collagen I messenger RNA (mRNA), although not collagen III mRNA, as well as increased total collagen protein synthesis [27]. Cardiac fibroblasts isolated from Lepr^db/db^ diabetic mice also showed increased collagen I mRNA and protein in culture [28]. Thus, the cardiac fibroblast response to “diabetic” conditions is to produce excess ECM, which aligns with the fibrosis observed in the diabetic heart in experimental models including the Lepr^db/db^ mouse [28]. Underscoring the clinical significance of this relationship, diabetes was associated with cardiac fibrosis in humans [29,30].

### 2.2. Conversion to a Myofibroblast Phenotype

Fibroblasts that produce more ECM often do so because of conversion to the more active myofibroblast phenotype; therefore, one would assume this to be the case for “diabetic” fibroblasts. Traditionally, myofibroblast conversion is assessed by an increase in α-smooth muscle actin (α-SMA) within the cell, making these cells contractile in nature and having the effect of contributing to the rearrangement and remodeling of the ECM [31]. Does HG cause a shift to the myofibroblast phenotype? The answer is not clear. Somewhat surprisingly, no differences were found, albeit at the mRNA level for α-SMA from isolated right-atrial cardiac fibroblasts obtained from diabetic and non-diabetic individuals without left-ventricular dysfunction, undergoing coronary artery bypass graft surgery [26]. Furthermore, Shamhart et al. [32] actually reported that less α-SMA was present in cultures of cardiac fibroblasts isolated from STZ type 1 diabetic rats, indicative of a reduction in myofibroblasts. This was supported by the finding of less α-SMA in the whole hearts of these STZ rats. On the other hand, cardiac fibroblasts isolated from Zucker obese diabetic rats induced slightly greater contraction of collagen gels than the lean counterpart, indicative of a more contractile myofibroblast phenotype [27]. This was supported by the finding of increased α-SMA mRNA in these Zucker cardiac fibroblasts. Similarly, isolated cardiac fibroblasts from normal neonatal rats showed increased α-SMA levels [33] and also generated greater gel contraction in response to HG conditions (25 mM) [23]. The increased contractile phenotype included increased active β1 integrin and increased α1 integrin [23]. Neonatal murine cardiac fibroblasts also demonstrate myofibroblast pheno-conversion in response to HG [25,34], and isolated adult rat cardiac fibroblasts underwent myofibroblast conversion (α-SMA) at a greater rate under HG conditions (25 mM) than normal glucose [22]. In this study, continual passaging led to even greater myofibroblast conversion. With such conflicting reports, it is unclear whether a HG environment promotes myofibroblast pheno-conversion. More important may be other culture conditions, including the length of time in culture, the substrate on which the cells are cultured, and number of passages. Comparing cells from different studies and different passage number may be akin to comparing apples and oranges when it comes to myofibroblast conversion. More in vivo identification of myofibroblasts may provide better information. What is clear, however, is that cardiac fibroblasts do not need to convert to the myofibroblast phenotype in order to produce excess ECM in response to HG.

### 2.3. Proliferation and Migration

Proliferation and increased migratory responses are also hallmark characteristics of more active fibroblasts. HG culture conditions induce rat and mouse cardiac fibroblast proliferation not due to increased osmolarity caused by excess glucose [22,25,34,35,36]. Cardiac fibroblasts isolated from the STZ rat model of type 1 diabetes were also shown to be more proliferative in culture than cells isolated from non-diabetic control mice [32], demonstrating similarities in fibroblast phenotype between type 1 and type 2 diabetes. These diabetic fibroblasts had reduced levels of the cell-cycle mediator p53. Since p53 is a tumor suppressor that utilizes p21 to inhibit cyclin-dependent kinase/cyclin activation and prevent cell-cycle progression, these data are supportive of cell-cycle progression and, hence, increased proliferation. Similarly, cardiac fibroblasts isolated from Zucker obese diabetic rats showed an increased proliferative rate compared to the lean controls when grown on a collagen I matrix [27]. However, reduced proliferation in response to HG (25 mM) was reported when naïve cardiac fibroblasts were grown on collagen [23]. This may indicate that the length of time of HG exposure is important, with cells from an in vivo diabetic state having prolonged exposure to HG versus cells from a normal in vivo state that are then treated with HG in vitro. In terms of human cardiac fibroblasts, there is some disagreement in the literature over proliferation responses, with one study finding increased proliferation for human cardiac fibroblasts obtained from biopsy of the right atrium of patients undergoing coronary bypass surgery or valve replacement (HG = 15 and 25 mmol/L) [37], and another study reporting no increase in proliferation for isolated right-atrial cardiac fibroblasts obtained from diabetic and non-diabetic individuals without left-ventricular dysfunction, undergoing coronary artery bypass graft surgery [26]. Some possible reasons for the differences in findings between these two studies include the following: (1) differences in gender composition of the individuals from which the fibroblasts were derived (four females and three males in Reference [37] versus one female and 11 males in Reference [26]); (2) the atrial fibroblasts were from non-diabetic individuals in Reference [37] and subsequently treated with HG media, whereas the atrial fibroblasts were derived from diabetic individuals and cultured in normal media in Reference [26]; (3) proliferation was assessed at one day in Reference [37], whereas the first time-point assessed in Reference [26] was at two days (final time-point = seven days). Thus, it is possible that there may be an acute proliferative response (one day) that then returns to normal by two days onward.

In terms of migration, under normal conditions, cardiac fibroblasts show a greater migratory tendency when cultured on collagen I and III over direct culture on plastic [22]. Addition of HG increases migration by roughly 10% on collagen I and III, as well as plastic. However, cardiac fibroblasts isolated from Zucker obese diabetic rats did not show increased migration in response to the scratch test [27]. An ECM substrate may be required to facilitate increased migration in response to HG.

Overall, there is clear conversion of cardiac fibroblasts to a phenotype that secretes excess amounts of collagen in response to a HG environment. This occurs whether cells are isolated from normal mice and then exposed to HG conditions, or whether cells are isolated from a diabetic environment, being either rodent models of type 1 or type 2 diabetes, or from diabetic humans. What is less clear is whether this phenotype change involves proliferation, myofibroblast conversion, and increased migratory responses. Primary cardiac fibroblasts change phenotype when cultured on plastic and with each subsequent passaging [22,38]. Thus, in vitro studies may report discrepant findings for proliferation, migration, and myofibroblast conversion due to differences in culture time and/or passage number. To our knowledge there are no studies that assessed the effects of HG on cardiac fibroblasts phenotype that cultured these cells on substrates that more closely mimic in vivo matrix stiffness characteristics. However, Fowlkes et al. [27] did culture cardiac fibroblasts isolated from diabetic male Zucker rats on collagen I matrix and compared the response to cells from lean controls. They demonstrated increased remodeling of collagen gels by diabetic fibroblasts, and that, while non-diabetic and diabetic fibroblasts showed an increase in proliferation on collagen I substrate compared to non-coated plates, there was no significant difference between the diabetic and non-diabetic cell types. Migration was not altered by culturing on collagen I for either diabetic or non-diabetic fibroblasts. It is also unclear whether the “diabetic” cardiac fibroblast phenotype is limited to one specific phenotype present at all stages of remodeling in the diabetic heart, or whether cardiac fibroblasts display a range of phenotypes depending on the stage of disease. Cardiac fibroblast phenotype is determined by numerous stimuli including mechanical force, growth factors, cytokines, and even the ECM [31]. These stimuli differ depending on the stage of remodeling and duration of the disease state and, thus, cardiac fibroblast phenotype may change depending on the stage of remodeling. This was recently demonstrated in myocardial ischemia where cardiac fibroblasts proliferate in the infarct/border zone region on days two to three, while converting to a myofibroblast phenotype on days three through 10 [39]. Thus, the myofibroblast phenotype persists for many days after cessation of proliferation, indicating the changing fibroblast phenotype as the remodeling process progresses. Additionally, mRNA expression profiles differed by time-point, furthering the concept of temporally regulated fibroblast phenotype. In vitro studies demonstrated that stiffness of the substrate on which isolated cardiac fibroblasts are cultured can also alter cell phenotype, with greater incorporation of α-SMA with increased substrate stiffness [40]. Interestingly, mechanical stretch, even when cardiac fibroblasts were cultured on substrates mimicking in vivo stiffness, induced upregulation of multiple genes for ECM proteins including collagen I, collagen III, and fibronectin [40]. While these examples are from ischemia and in vitro studies, it seems likely that mechanical and temporal changes in fibroblast phenotype in diabetes occur. This may account for the unclear findings regarding myofibroblast conversion and proliferation. However, to our knowledge, there are no studies that investigated the cardiac fibroblast response to HG using substrates that more closely mimic the in vivo environment. Regardless, however, the main effect of HG is the excess production of ECM by cardiac fibroblasts that defines fibrosis.

## 3. Pathways Mediating Fibroblast Phenotype and Function

Studies investigating the response of cardiac fibroblasts to HG identified a number of pathways modulated by HG that underlie the shift to a pro-fibrotic cardiac fibroblast phenotype.

### 3.1. Renin Angiotensin System

HG conditions increase the amount of angiotensin II (ang II) type 1 (AT_1_) receptor on rat cardiac fibroblasts, with AT_1_ receptor blockade able to oppose the increase collagen production caused by HG [17,19]. Interestingly, there was no additive effect of ang II in addition to HG in terms of collagen production [17], reinforcing that HG induces ang II production by fibroblasts to induce collagen production. To this end, using confocal microscopy, Singh et al. [19] identified perinuclear and nuclear distribution of intracellular ang II. Angiotensinogen, renin, and angiotensin-converting enzyme (ACE) could all be detected in isolated cardiac fibroblasts, with renin levels increasing upon stimulation with HG. Both renin and ACE inhibition prevented the increase in ang II synthesis in response to HG, as well as preventing collagen synthesis by these cardiac fibroblasts [19]. Thus, there appears to be a local renin angiotensin system activated in cardiac fibroblasts that mediates the pro-fibrotic response to HG.

### 3.2. Transforming Growth Factor-β

Isolated cardiac fibroblasts from Lepr^db/db^ mice produced substantially increased amounts of transforming growth factor (TGF)-β in culture compared to fibroblasts from non-diabetic control mice [28]. HG stimulation of neonatal rat cardiac fibroblasts also induced an increase in the amount of active TGF-β without increasing the overall pool of available TGF-β [18]. Isolated rat cardiac fibroblasts produced excess TGF-β1, Smad2/3, and decreased Smad7 in response to HG conditions (25 mM) [21], again indicative of an activated TGF-β pathway, with Smad3 contributing to myofibroblast conversion in cardiac fibroblasts isolated from STZ diabetic rats [41]. Similarly, TGF-β1 and Smad2/3 activation were also upregulated in isolated human cardiac fibroblasts by HG (30 mM) [42], indicating the translational significance of this pathway. Incubation with the inhibitor of TGF-β production, tranilast, reduced [^3^H]-proline incorporation by cardiac fibroblasts under HG conditions, indicative of a role for TGF-β1 in fibroblast proliferation [43]. Furthermore, treatment of mouse neonatal cardiac fibroblasts with a neutralizing antibody against TGF-β prevented the upregulation of periostin and α-SMA that occurs in response to HG [34], indicating a role for TGF-β in ECM production and proliferation. At the whole-animal level, tranilast was also able to prevent fibrosis in a Ren-2 rat STZ model of diabetes [43], seemingly by reduction of TGF-β1 levels and subsequent Smad2 activation. Thrombospondin-1 (TSP1) is a major activator of latent TGF-β to stimulate increased TGF-β activity. HG conditions lead to an upregulation of TSP1 by isolated neonatal rat cardiac fibroblasts [18], with inhibiting peptides, as well as a neutralizing antibody, demonstrating that TSP1 was involved in TGF-β activation in response to HG [18]. Interestingly, the AT_1_ receptor antagonist losartan did not alter HG induction of TGF-β, suggesting that the aforementioned effects of ang II on fibroblast function under HG conditions do not involve TGF-β [18]. However, this is unclear, since inhibition of renin and ACE did reduce TGF-β levels in neonatal cardiac fibroblasts [19]. Regardless, increased active TGF-β appears to be a consistent response to HG, resulting in increased ECM production, proliferation, and myofibroblast transition.

### 3.3. Kinases

Extracellular signal-regulated kinase (ERK)1/2 is a common mediator for HG-stimulated proliferation in cells [44]. Interestingly, Shamhart et al. [22] found that ERK1/2 phosphorylation was initially decreased (20 min) in adult rat cardiac fibroblasts exposed to HG (25 mM) before increasing back to basal levels by 24 h. Conversely to these rat adult cardiac fibroblasts, ERK1/2 phosphorylation and kinase activity were increased in isolated adult mouse cardiac fibroblasts in response to HG [24]. Similarly, mouse neonatal cardiac fibroblasts also showed upregulated ERK activation in response to HG, with inhibition of ERK activity with U0126 able to prevent the upregulation of TGF-β and reduce the proliferative response to HG [34]. This established a causal role for ERK in mediating the pro-fibrotic phenotype induced by HG, but it may also reflect species differences in HG-induced signaling. In isolated human cardiac fibroblasts adenosine monophosphate-activated protein kinase (AMPK) and p38 activation were increased in response to HG (30 mM) [42]. The AMPK inhibitor, compound C, was able to reduce collagen production by these human cardiac fibroblasts, confirming the role of AMPK in this process. Phospho-protein kinase B (Akt) was found to be increased by HG (30 mM) in neonatal murine cardiac fibroblasts [25]. Phosphoinositide 3 (PI3) kinase levels were increased by HG (25 mM) in isolated adult murine cardiac fibroblasts [24], as was phosphorylated Akt. Akt activation was found to be dependent on PI3 kinase and then subsequently activated ERK1/2. Thus, HG conditions cause activation of a number of kinases potentially including ERK1/2, p38, AMPK, and Akt. These kinases appear to be important in promoting a pro-fibrotic fibroblast phenotype in response to HG, but they may show species-specific differences.

### 3.4. Extracellular Matrix Glycation/Advanced Glycation End Products/Receptor for Advanced Glycation End Products/*O*-GlcNAcylation

#### 3.4.1. Extracellular Matrix Glycation

Hyperglycemia results in increased glycation of ECM proteins with subsequent formation of advanced glycation end products (AGEs). This is a non-enzymatic process, which can result in alterations in cell adhesion to the ECM [45], as well as changes to the biochemical functions of the ECM [46], thereby causing irreversible cross-linking between ECM proteins and increased stiffness of the heart [47,48]. AGEs can accumulate in diabetic tissue up to 14 times faster than in normal tissue [49]. An interesting study by Yuen et al. [50] tested the response of human cardiac fibroblasts cultured on collagen gels that underwent prior glycation by methylgloxal, a glucose metabolite. Collagen gel contraction was greater for fibroblasts on gels that underwent glycation, as was myofibroblast conversion. Myofibroblast conversion could be prevented by aminoguanidine that inhibits collagen cross-linking, indicating that glycation-induced cross-linking of collagen was a stimulus for myofibroblast conversion. Migration was also increased on glycated collagen. The α_2_β_1_ integrin complex is essential for fibroblast interactions with collagen. Interestingly, however, the β_1_ integrin sub-unit was inhibited under glycated collagen conditions. Microarray analysis of human cardiac fibroblasts cultured on glycated collagen showed a 2.1-fold increase in expression of α_11_ integrin, while α_1_, α_2_, and α_10_ integrins were unchanged [41]. Thus, ECM alterations caused by glycation have selective effects on integrins. Isolation of cardiac fibroblasts from STZ diabetic rats found that these fibroblasts had higher levels of α_11_ integrin and α-SMA. This was also true of whole-heart lysates. Interestingly, overexpression of α_11_ integrin (90-fold) in non-diabetic mice causes cardiac fibrosis when examined in mice six and 12 months of age, with some overlap of areas with α_11_ integrin and collagen deposition [51]. TGF-β2 and TGF-β3, but not TGF-β1, mRNA was upregulated in these α_11_ integrin overexpressing mice. When α_11_ integrin was knocked down by small interfering RNA (siRNA) in human cardiac fibroblasts, there was a marked reduction of α-SMA, indicating a critical role for α_11_ integrin in promoting a myofibroblast phenotype in response to “diabetic” conditions. α_11_ integrin and subsequent myofibroblast conversion were regulated by Smad3, linking back to a possible role of TGF-β1 in the regulation of myofibroblast phenotype and integrin function in fibroblasts under HG conditions.

#### 3.4.2. Advanced Glycation End Products/Receptor for Advanced Glycation End Products

A study by Tang et al. [20] demonstrated the ability of AGEs to have direct pro-fibrotic effects on adult rat cardiac fibroblasts. In this study, AGEs, generated by incubating bovine serum albumin in 50 mmol/L d-glucose at 37 °C for eight weeks, were shown to upregulate collagen I gene expression and also protein levels in these cardiac fibroblasts. Somewhat surprisingly, collagen III mRNA and protein were downregulated by AGEs. Regulation of collagen I in response to AGEs involved both p38 and ERK, with a greater contribution from ERK. The downregulation of collagen III appeared to solely involve ERK. c-Jun N-terminal kinase (JNK) inhibition did not alter regulation of either collagen. This is interesting because collagen I is described as being responsible for the tensile strength in the heart, whereas collagen III is considered more compliant. These data would be consistent with AGEs contributing to increased stiffness in the diabetic heart. Interestingly, mechanical stiffening of gels (from 8 kPa to 30 kPa) on which isolated cardiac fibroblasts were cultured under normal glucose conditions caused upregulation of col1a1 and col2a2 mRNA, while col3a1 mRNA was reduced [40]. Similarly, in the stiffening heart post myocardial infarction, there is a shift toward increased collagen I (57% vs. 16% in control) and a loss of collagen III (0% vs. 2% in control) [52]. Thus, mechanical ECM changes may also contribute to collagen upregulation, including in diabetes.

It appears that many of the actions of AGEs at the cardiac fibroblast may be mediated by ang II. AGE-induced cardiac fibroblast proliferation could be prevented by the AT_1_ receptor antagonist, losartan [49]. Incorporation of proline by cardiac fibroblasts, as a marker of collagen production, was also increased by AGEs and, again, this could be prevented by losartan. AGE-induced translocation of the p65 sub-unit of nuclear factor kappa B (NF-κB) to the nucleus could be prevented by AT_1_ receptor blockade [49]. AGEs can have effects such as inducing pro-fibrotic pathways through receptors including the receptor for advanced glycation end products (RAGE). RAGE was found to be upregulated in diabetic cardiac fibroblasts [28], and it can induce cardiac fibroblast proliferation [49]; however, the extent to which the effects of AGEs are mediated by RAGE is currently unclear. Interestingly, the increased levels of RAGE induced by AGEs could also be prevented by losartan. Thus, the renin angiotensin system may mediate a pro-fibrotic fibroblast phenotype via an increase in RAGE, allowing for increased effects of AGEs.

#### 3.4.3. *O*-GlcNAcylation

*O*-GlcNAcylation is an alternative method for the addition of glucose adducts via an enzymatic process where a β-*N*-acetylglucosamine (*O*-GlcNAc) molecule is attached to proteins via *O*-linkage by the *N*-acetylglucosamine transferase to serine and threonine residues [53]. *O*-GlcNAcylation is a way of modifying protein function. HG conditions cause increased *O*-GlcNAcylation of Sp1 (a transcription factor), as well as arginase I and II. Greater amounts of *O*-GlcNAcylated Sp1 were bound to the collagen I gene promoter under HG conditions. The functional significance of *O*-GlcNAcylation was demonstrated by adenovirus administration of *N*-acetylglucosaminidase, which removes *O*-GlcNAc moieties from proteins. This prevented TGF-β1 and Smad2/3 upregulation, as well as Sp1 and arginase *O*-GlcNAcylation, ultimately reducing collagen production [21]. Thus, TGF-β1 is at least one mechanism via which *O*-GlcNAcylation induces a pro-fibrotic phenotype in cardiac fibroblasts.

### 3.5. Cytokines

Cytokines play a significant role in contributing to the activation of cardiac fibroblasts and collagen deposition in diabetic cardiomyopathy both in vivo and in vitro, with some cytokines being pro-fibrotic and some being anti-fibrotic.

#### 3.5.1. Interleukin-6

Incubation of isolated neonatal mouse cardiac fibroblasts in HG (25 mM) resulted in increased interleukin (IL)-6 production by these cells [54]. Addition of IL-6 alone increased fibroblast proliferation and collagen mRNA expression, with deletion of IL-6 preventing cell proliferation and upregulated collagen gene expression in response to HG. This likely involved the reduction of TGF-β1. Interestingly, deletion of IL-6 in these neonatal cardiac fibroblasts allowed for a dramatic increase in miR-29, whilst addition of IL-6 downregulated miR-29. Inhibition of miR-29 in the presence of IL-6 allowed upregulation of TGF-β, collagen I, and collagen III mRNA, whilst overexpression of miR-29 in the presence of IL-6 downregulated these parameters. Thus, under HG conditions, the increase of IL-6 appears to upregulate collagen genes via induction of TGF-β, with this pathway allowed to persist by IL-6 downregulation of miR-29. However, the relevancy of this to human cardiac fibroblasts is unclear. No differences were found in mRNA levels for IL-6, as well as IL-1β and IL-8, for isolated right-atrial cardiac fibroblasts obtained from diabetic and non-diabetic individuals without left-ventricular dysfunction, undergoing coronary artery bypass graft surgery [26].

#### 3.5.2. Interleukin-17

IL-17 was increased in the STZ mouse heart, with deletion of IL-17 preventing increases in TGF-β1, α-SMA, collagen I, and collagen III mRNA, as well as restoring normal systolic and diastolic function [55]. Increasing concentrations of glucose (up to 25 mM) showed a concentration-dependent induction of IL-17 by isolated adult murine cardiac fibroblasts [24]. IL-17 production could be blocked by both the inhibitor of RNA polymerase II, actinomycin D, and the protein synthesis inhibitor cycloheximide, indicating that IL-17 production is regulated at both the transcriptional and the post-transcriptional levels. HG conditions were found to induce IL-17 production by the PI3 kinase/Akt/ERK pathway. Importantly, IL-17 was involved in increased collagen production in cardiac fibroblasts in response to HG, with both subunits of the IL-17RA and IL-17RC heterodimer complex being important to mediating this response. IL-17 was also upregulated by HG (25 mM) in neonatal mouse cardiac fibroblasts, with deletion of IL-17 preventing the HG-induced increase in cell viability, as well as mRNA levels of TGF-β1, α-SMA, collagen I, and collagen III [55].

#### 3.5.3. Interleukin-1β

The pro-inflammatory cytokine, IL-1β, was upregulated in isolated rat neonatal cardiac fibroblasts treated with HG (30 mM and 50 mM) [56]. This coincided with the increased levels of NF-κB and the upregulation of the TGF-β1/Smad pathway, which ultimately led to excessive production of collagen type I and type III. The HG environment caused the phosphorylation of NF-κB and promoted the production of IL-1β in cardiac fibroblasts. The inhibition of NF-κB led to not only a decrease in IL-1β levels but also a decrease in cell viability, prompting the role of NF-κB in HG-induced cardiac inflammation.

#### 3.5.4. Interleukin-33

Another cytokine produced by cardiac fibroblasts is IL-33. This cytokine belongs to the IL-1 family, and it was shown to be anti-fibrotic and anti-hypertrophic in a pressure overload model [57,58]. In the STZ model of diabetes, long-term administration of exogenous IL-33 (5 weeks) attenuated collagen type I deposition, as well as improved cardiac function [59]. This appears to involve direct effects on cardiac fibroblasts since HG challenge (30 mM) significantly reduced IL-33 levels in isolated mouse cardiac fibroblasts. Interestingly, IL-33 was further reduced in the presence of high-mobility group box 1 (HMGB1), through both exogenous administration and co-culture with cardiomyocytes, which is the major source of HMGB1 in an HG environment, leading to increased collagen type I [59]. HMGB1 bound to toll-like receptor 4 (TLR4) on the fibroblasts, in turn downregulating IL-33 and increasing collagen production [59]. However, the overexpression of intracellular IL-33 abrogated the HG/HMGB1-induced collagen type 1 production.

### 3.6. Matrix Metalloproteinases

HG conditions caused reduced total matrix metalloproteinase (MMP) activity in the media from adult rat cardiac fibroblasts [17]. This could contribute to the accumulation of collagen that occurs in response to HG, since MMPs can degrade ECM. However, MMP-2 and MMP-9 were reported to increase in neonatal rat cardiac fibroblasts in response to HG conditions [35]. Similarly, MMP-2 and MMP-9 protein levels and activity were increased by HG in murine neonatal cardiac fibroblasts [25]. Thus, while the overall pool of MMP activity may be downregulated in response to HG, specific MMPs are likely upregulated (e.g., MMP-2 and MMP-9) and could have a pro-fibrotic role. MMP-2 and tissue inhibitor of metalloproteinases (TIMP)-2 were both increased in cardiac fibroblasts isolated from Lepr^db/db^ diabetic mice [28]. Alternatively, no differences in MMP-2 or MMP-3 were found, albeit at the mRNA level for isolated right-atrial cardiac fibroblasts obtained from diabetic and non-diabetic individuals without left-ventricular dysfunction, undergoing coronary artery bypass graft surgery [26].

### 3.7. Non-Coding RNA

Non-coding RNA are RNA molecules that are not translated into protein, but which function to regulate gene expression and transcription. The long non-coding RNA AK081284, is increased in mouse cardiac fibroblasts under HG conditions (25 mM) [55]. This increase was mediated by IL-7. The siRNA knockdown of AK081284 normalized mRNA levels for TGF-β1, α-SMA, collagen I, and collagen III, indicating a probable role in promoting a pro-fibrotic phenotype for cardiac fibroblasts in response to HG.

MicroRNAs (miRNAs) are non-coding, small RNAs that regulate gene expression at the post-transcriptional level. miR-155 is important to a wide array of biological processes, including immunity, inflammation, autophagy, cancer, and fibrosis [60,61,62,63]. Deletion of miR-155 in mice reduced cardiac fibrosis in response to injection with STZ [64]. Isolated cardiac fibroblasts displayed increased levels of miR-155 under HG conditions (40 mM) with inhibition of miR-155 capable of reducing collagen production by these cells. Increased activation of miR-155 further increased collagen production. Inhibition of miR-155 was found to inhibit TGF-β1 levels, as well as subsequent induction of Smad2/3 activation, providing a pathway for the pro-fibrotic effects of miR-155.

### 3.8. Myocyte Enhancer Factor 2

Myocyte enhancer factor 2 (MEF2) is a DNA-binding transcription factor capable of activating muscle-, growth factor-, and stress-induced genes [65]. MEF2A and MEF2D are the primary MEF2s expressed in the adult heart. MEF2A was upregulated by HG (33 mM) in neonatal murine cardiac fibroblasts [25]. Cellular proliferation, increased migratory properties, and myofibroblast conversion induced by HG could be prevented by knockdown of MEF2A by short hairpin RNA (shRNA). However, MEF2 did not contribute to apoptosis of these cells. MEF2A was important for upregulation of MMP-2 and MMP-9 protein, but it did not regulate TMP1 or TIMP2. Accordingly, MMP-2 and MMP-9 activity was attenuated by MEF2A knockdown. Ultimately, MEF2A was also important to the increase in collagen production in response to HG. From a signaling point of view, MEF2A was responsible for upregulated Akt [25], which is part of a signaling cascade resulting in excess collagen production by cardiac fibroblasts in response to HG [24]. MEF2A was also responsible for upregulation of TGF-β1 and subsequent Smad2 and Smad3 signaling, which also contribute to collagen production by cardiac fibroblasts in response to HG. In vivo, MEF2A mRNA and protein were significantly increased in the STZ model of type 1 diabetes, with MEF2A knockdown attenuating cardiac fibrosis with improved cardiac function [25].

### 3.9. Oxidative Stress

Diabetes induces oxidative stress that contributes to cardiac fibrosis [66]. An HG environment promotes increased NO production by adult cardiac fibroblasts isolated from female Sprague-Dawley rats [67]. Nitric oxide (NO) is generally considered to be protective when it comes to the cardiovascular system [68]; however, NO can have adverse effects when it reacts with superoxide, another free radical, to produce peroxynitrite [69]. That this increased NO was due to upregulated inducible nitric oxide synthase (iNOS) suggests that NO may be linked to peroxynitrite production. Thus, elevated NO likely represents increased oxidative stress in a HG environment. Further supporting a contribution for oxidative stress to fibroblast phenotype in diabetes, the free radical hydrogen peroxide (H_2_O_2_) can induce proliferation in rabbit atrial fibroblasts, and oxidative stress contributes to HG-induced proliferation in these cells, since the reduced nicotinamide adenine dinucleotide phosphate inhibitor, apocynin, could inhibit proliferation [70]. HG induced p38 activation, which was prevented by apocynin inhibition of oxidative stress. MMP-9 upregulation was also mediated by oxidative stress. In isolated rat cardiac fibroblasts, the oxidative stress inhibitor *N*-acetylcysteine prevented HG-induced (25 mM) proliferation, connective tissue growth factor upregulation, and increased collagen expression [66]. Thus, oxidative stress is an important component of the response to HG that results in a change in cardiac fibroblasts to a more active phenotype.

### 3.10. Nucleotide Oligomerization-Binding Domain 1

Nucleotide oligomerization-binding domain 1 (NOD1) is an innate immune receptor that is involved in the pathogenesis of diabetes in various organs, including the heart. Although increased NOD-1 is associated increased levels of NF-κB and apoptosis in diabetic mice, its underlying contribution still remains poorly understood [71]. NOD1 is found in cardiac fibroblasts, with levels increased in cardiac fibroblasts from Lepr^db/db^ mice compared to non-diabetic controls [72]. These cells showed concurrent Smad activation, indicative of potential TGF-β upregulation, as well as greater translocation of the p65 subunit of NF-κB to the nucleus, indicative of NF-κB activation. Accordingly, NOD1 inhibition in vivo decreased protein levels of p-Smad2/3, as well as reduced mRNA levels of collagen I in isolated Lepr^db/db^ mouse cardiac fibroblasts. Similar results were also reported for NF-κB inhibition. Critically, fibrosis was reduced in Lepr^db/db^ mice by inhibition of NOD1. Thus, NOD1 may be an important molecule in cardiac fibroblasts under HG conditions by inducing NF-κB activation, leading to conversion to a pro-fibrotic phenotype.

### 3.11. Methyl CpG-Binding Protein 2

Methyl CpG-binding protein 2 (MeCP2) is a transcription repressor that acts by binding to the methylated sequences of target genes, leading to gene silencing. MeCP2 is upregulated by HG. The siRNA knockdown of MeCP2 reduced isolated neonatal rat cardiac fibroblast proliferation, collagen production, and myofibroblast conversion in response to HG (33.3 mM), indicating a potential role for MeCP2 in the induction of a pro-fibrotic phenotype in “diabetic” fibroblasts [33]. Knockdown of MeCP2 allowed Ras association domain family 1 isoform A (RASSF1A) to increase, which induces methylation [33]. The siRNA knockdown of RASSF1A caused increased fibroblast proliferation. Together, these findings indicate that induction of MeCP2 by HG allows fibroblast conversion to a pro-fibrotic phenotype to proceed by downregulating RASSF1A, which removes the protective methylation of important genes.

### 3.12. Protease Activated Receptor 4

Protease activated receptor 4 (PAR-4) is an innate G protein-coupled receptor that is activated by serine proteases such as thrombin [73]. Diabetic STZ mice show increased mRNA levels of protease activated receptor 4 (PAR-4), and mouse cardiac fibroblasts expressed PAR-4, with levels upregulated by HG (25 mM) [74]. Interestingly, mRNA levels of IL-6, α-SMA, and periostin increased in response to thrombin, a PAR-4 agonist. The response was abrogated on PAR-4^−/−^ cells. The migratory response to HG was also slightly increased when PAR-4 was activated. The significance of PAR-4 in the diabetic heart is uncertain since it is not clear whether thrombin is likely to activate PAR-4 in the heart.

## 4. Approaches to Oppose High Glucose-Induced Pro-Fibrotic Cardiac Fibroblast Phenotype

Several studies investigated the ability of various compounds to oppose the effects of HG to induce a pro-fibrotic phenotype in cardiac fibroblasts. While not extensive, these studies do provide some hope for approaches that may reduce fibrosis in the diabetic heart. The relatively small number of studies highlights the need for more focus in this area.

### 4.1. Relaxin

Relaxin is a 6-kDa polypeptide pregnancy-related hormone also expressed in non-reproductive organs such as the heart. It can prevent fibrosis in mouse models of hypertension, as well as reduce collagen deposition in ang II- and TGF-β-treated rat neonatal cardiac fibroblasts [75]. Relaxin 1 mRNA was initially increased in neonatal rat cardiac fibroblasts under HG conditions (33 mM), whereas relaxin 3 mRNA was downregulated [76]. The relaxin receptor RXFP1, which is the receptor of interest in the cardiovascular system, was downregulated at the mRNA level, whilst RXFP3 was upregulated. Importantly, relaxin was able to reduce MMP-2 and MMP-9 levels in isolated cardiac fibroblasts, as well as the proliferative response of neonatal rat cardiac fibroblasts incubated with high glucose (25 mM) [35].

### 4.2. Resveratrol

Resveratrol is a phytoalexin found largely in the skins of red grapes and other fruits. Resveratrol was able to prevent proliferation of rat neonatal cardiac fibroblasts under HG conditions (25 mM) [36]. Excess production of both collagen I and III could also be prevented by resveratrol by counteracting the HG-induced upregulation of TGF-β1 [36]. Another study using neonatal mouse cardiac fibroblasts also reported the ability of resveratrol to oppose proliferation and myofibroblast conversion, as well as ROS production under HG conditions (25.5 mM) [34]. Increased ERK activation and TGF-β in response to HG could be ameliorated by resveratrol. Similarly, pre-treatment of isolated adult mouse cardiac fibroblasts with resveratrol could prevent increased collagen production by cardiac fibroblasts in response to HG (25 mM). In this study, this effect of resveratrol was due to inhibition of PI3K activity induced by HG [24], as well as Akt kinase activity, ERK1/2 activity, and IL-17 production. These findings appear to be of relevance in vivo since resveratrol reduced periostin, TGF-β, and activated ERK levels, and prevented cardiac fibrosis in the STZ mouse model of diabetes [34]. Thus, resveratrol is linked to HG conditions via a number of pathways.

### 4.3. Curcumin

Curcumin is a naturally occurring polyphenol that can be extracted from the plant *Curcuma longa* L. [77]. Curcumin substantially reduced cardiac fibrosis in the STZ rat model of diabetes, seemingly by reducing AGEs, Akt, and TGF-β1 levels, as well as subsequent activation of the Smad2/3 pathway [42,77]. In isolated human cardiac fibroblasts, curcumin was shown to inhibit the increase in TGF-β1 and Smad2/3 activation that occurs in response to HG (30 mM), as well as the subsequent increase in collagen production [42]. Curcumin could prevent the activation of AMPK and p38 in these human fibroblasts.

### 4.4. Matrine

The active molecule of the Chinese herb *Sophora alopecuroides* L. is known as matrine (C_15_H_24_N_2_O). Matrine attenuated fibrosis in a rat STZ-induced diabetes model [78]. Incubation of isolated neonatal rat cardiac fibroblasts with HG (25 mM) caused upregulation of ATF6p50, calreticulin, fibronectin, and collagen I. Matrine was able to concentration-dependently reduce the levels of each of these molecules. The relevancy of this in vivo was indicated by the ability of matrine to also attenuate production of these molecules in the STZ diabetic rat heart. Matrine could also concentration-dependently reduce nuclear translocation of NFATc1 in cardiac fibroblasts.

### 4.5. Tanshinone

Tanshinone IIA is an extract from the Chinese herb danshen. Tanshinone IIA was able to concentration-dependently oppose proliferation and proline incorporation by neonatal rat cardiac fibroblasts in response to HG (25 mM) [79]. Tanshinone IIA was able to inhibit HG-induced production of ROS and TGF-β1, since TGF-β1 mRNA and protein levels were decreased by tanshinone iia, as was Smad2/3 phosphorylation.

### 4.6. Trimetazidine

Trimetazidine is an anti-anginal agent that selectively inhibits the activity of mitochondrial long-chain 3-ketoacyl-CoA thiolase to cause inhibition of free-fatty-acid oxidation and promotion of glucose oxidation [80]. Trimetazidine was able to oppose increased collagen synthesis by isolated neonatal rat cardiac fibroblasts in response to HG, likely via downregulation of connective tissue growth factor and oxidative stress [81]. Importantly, trimetazidine reduced cardiac fibrosis in the STZ model of type 1 diabetes.

## 5. Limitations of the Literature and Future Directions

From the available in vitro studies, the HG environment activates various molecular pathways in cardiac fibroblasts to induce excessive collagen deposition (Figure 1). Interestingly, these various pathways, more often than not, appear to culminate in an overall increase in active TGF-β1. What is not fully clear is the extent to which the many pathways involved act via RAGE or are induced by RAGE activation. Future studies should aim to further understand the intricacy of these pathways, how they interact, and the contribution of RAGE to the activation of these pathways. Furthermore, a greater emphasis needs to be placed on identifying approaches that act at the fibroblast to oppose the pro-fibrotic phenotype induced by HG. While some potential therapies to target the “diabetic” fibroblast were identified (Figure 2), these only underwent initial investigation with only a very rudimentary understanding of how they alter pro-fibrotic pathways induced in cardiac fibroblasts by HG (Figure 2).

Most of the studies included in this review investigated the response of isolated cardiac fibroblasts to HG. This approach was a necessity due to the lack of specific markers for fibroblasts that allow for their manipulation in vivo; however, this represents a major challenge in clinically translating the knowledge from current studies. This is likely to change due to the identification of *Tcf21* as a specific marker for fibroblasts, which now allows for manipulation of fibroblasts in vivo with the development of a mouse with tamoxifen-inducible Cre recombinase linked to *Tcf21* [82,83]. This will allow for manipulation of specific molecules within fibroblasts. However, to date, no studies specifically manipulated fibroblasts in vivo in the setting of diabetes. This is also critical to identifying potentially translatable findings because in vivo interrogation of fibroblasts takes into account the complex responses of fibroblasts to hemodynamic effects, cell-to-cell interactions, and even organ system interactions. While all fibroblasts have certain key shared features, fibroblasts from different tissues of the body are also considered to be unique from one another, which is a product of their local microenvironment. Yet, at the same time, they can be influenced by events in tissues remote to their location. It is important to remember that, in all likelihood, interactions between the heart and other organs occur in the diabetic state in vivo that could modulate cardiac fibroblast function and that are not captured in isolated fibroblast experiments. This was demonstrated by Trevisan et al. [84], who isolated dermal fibroblasts from diabetic individuals with nephropathy and compared collagen production with dermal fibroblasts from diabetic individuals without nephropathy, as well as with dermal fibroblasts from healthy individuals. Intriguingly, dermal fibroblasts from individuals with nephropathy synthesized significantly higher amounts of collagen compared to dermal fibroblasts from diabetic individuals without nephropathy and healthy controls. Collagen degradation rate was no different between groups, indicating increased collagen synthesis by the nephropathy group. This shows that disease in one organ (in this case, the kidney) could influence fibroblast phenotype in another organ (in this case, the skin). Thus, the ability to manipulate the phenotype of diabetic cardiac fibroblasts in vivo, as well as pathways and molecules identified by in vitro studies, will be a critical next step forward in our understanding of cardiac fibroblast function in the diabetic heart.

## 6. Conclusions

Conditions that mimic diabetes, such as a HG environment, increase ECM production. What is less clear is whether the HG environment promotes cardiac fibroblast proliferation or the transition of these fibroblasts to a myofibroblast phenotype, or whether it enhances their migratory response. The HG environment activates various molecules to induce excessive collagen deposition by cardiac fibroblasts, including the renin angiotensin system, multiple kinases, cytokines, MMPs, non-coding RNA molecules, oxidative stress, and other signaling molecules (Figure 1). Interestingly, these various pathways, more often than not, appear to culminate in an overall increase in active TGF-β1.

## Figures and Tables

**Figure 1 ijms-21-00970-f001:**
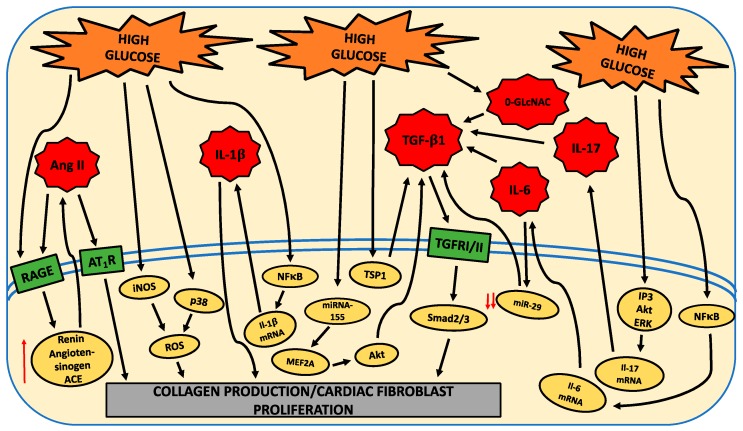
Schematic indicating molecules and intracellular pathways induced by high glucose to promote a pro-fibrotic cardiac fibroblast phenotype. High glucose (HG) causes the activation of intracellular pathways (yellow ovals) that directly induces a pro-fibrotic phenotype in cardiac fibroblasts or, alternatively, upregulates molecules (red stars), which in turn activate intracellular pathways that induce a pro-fibrotic phenotype in cardiac fibroblasts. Red ↑ = increased levels, red ↓↓ = dramatically decreased.

**Figure 2 ijms-21-00970-f002:**
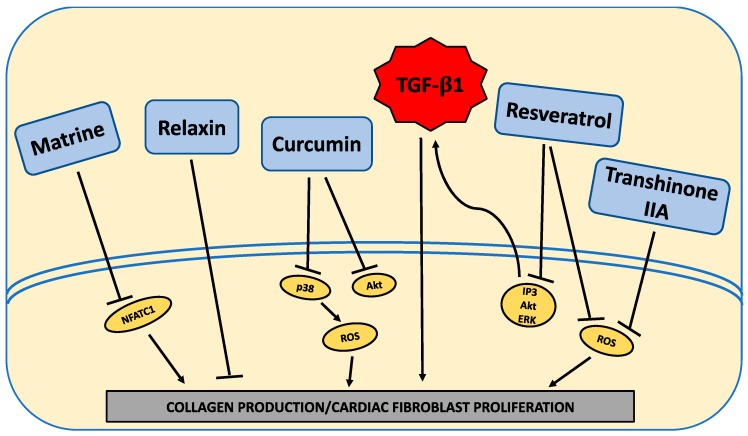
Schematic indicating compounds that oppose the high-glucose-promoted pro-fibrotic actions on cardiac fibroblasts. Intracellular signaling pathways (yellow ovals) are induced in cardiac fibroblasts by high glucose to induce a pro-fibrotic phenotype. Anti-fibrotic compounds (blue rectangles) inhibit various intracellular signaling pathways to oppose the induction of a pro-fibrotic phenotype in cardiac fibroblasts under high glucose conditions.

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
