# Peer review of "The Diabetic Cardiac Fibroblast: Mechanisms Underlying Phenotype and Function"

_ijms, 2020, doi:10.3390/ijms21030970_

Round 1
Reviewer 1 Report
This review by Levick and Widiapradja is well-organized and thorough. It reviews the current knowledge of cardiac fibroblast activity and phenotype in diabetes, as well as diabetes-induced pro-fibrotic signaling pathways. It also summarizes the limited number of studies investigating anti-fibrotic effects on cardiac fibroblasts.
The following points could improve the paper further:
The authors should discuss whether the diabetic fibroblast phenotype is limited to one specific phenotype for all stages and degrees/types of diabetes? Cardiac fibroblasts display a range of phenotypes in other cardiac diseases, depending on the stage of disease and various mechanical cues (e.g. Fu et al., JCI 2018, Herum et al., JCM 2017). The stage of disease could potentially also explain conflicting results in findings regarding the myofibroblast phenotype in cardiac fibroblasts isolated from diabetic hearts (section 3 page 2).
The authors should also discuss the limitations of in vitro studies of cardiac fibroblasts. Primary cardiac fibroblasts change phenotype during culturing on plastic. Thus, there may be phenotypic differences in the various in vitro studies depending on culturing time. Are there any studies investigating HG effects on cardiac fibroblasts on hydrogels with in vivo-like stiffness?
Please summarize the part about proliferation on page 3, line 117. In the conclusion the authors state that HG increases proliferation. This was not clear when reading the part about proliferation.
Page 4, line 156. Tranilast can inhibit several other pathways than TGFb. Has it been verified to specifically inihibit TGFb? If so the reference should be given. Tranilast can inhibit effects caused by TGFb, but the mechanism is not known.
If possible, section 5.2 and 5.3 would be improved by including a final sentence summarizing each section.
The authors should discuss and include the paper by Romaine et al., describing fibrosis in integrin alpha 11 overexpressing mice. (Romaine et al., Acta Physiol, 2018)
Page 5, line 217. The authors could discuss other studies finding opposite regulation of collagen I and III, which has also been observed in vivo (Sullivan et al., Stem Cell Res 2014) and in vitro (Herum et al., MBoC 2017) in response to matrix stiffening.
Several places in the text, it is stated that “this was not due to osmotic effects of glucose”. Does this mean that in many studies, the osmotic effects of glucose are not controlled?
Sections 5.9 – 5.11 should be explained more clearly to help the reader understand the important messages. E.g. line 336-338, line 351-352 (“the same was true for … , indicating…).
Although it is good that the limitations of isolated cardiac fibroblast experiments is mentioned, the last part of the conclusion is not very summarizing of the review. The authors could add a separate section mentioning these limitations since most of the reviewed literature is based on in vitro cardiac fibroblast experiments. This sections could also include the aforementioned limitations of cardiac fibroblast spontaneous activation and differentiation in vitro.
Since it is not know whether high glucose induces a myofibroblast phenotype, Figure 1 and Figure 2 should read "cardiac fibroblast proliferation" instead of "myofibroblast proliferation". Figure 2 could be compacted to reduce empty space.
Minor comments:
Page 2, line 45 "pro-phenotype" should be "pro-fibrotic phenotype"
Page 2, line55. Was collagen I production increased in diabetic- compared to non-diabetic patients? Please clarify.
Page 3, line 95. Should it read “This difference was greater”?
Page 3, line 110. Please specify whether the patients studies in ref 36 had left ventricular dysfunction
Page 3 line 122. “This was despite up-regulated beta1 integrin…”: Increased adhesion will often lead to less migration.
Page 4 line165. Please state what losartan is for non-clinical readers.
Page 4 line 180. Define abbreviation AMPK. Also in list of abbreviations.
Page 6 line 281. Sentence starting “In the STZ model…” is confusing. Please rephrase.
Page 6 Add reference for line 283-289
Page 7 line 292. Are these adult rat cardiac fibroblasts compared to the neonatal rat fibroblasts mentioned in line 295? Please state adult/neonatal in line 292.
Relaxin has been shown to act through AT2 receptors. Are there any studies showing anti-fibrotic effect of AT2 activation in diabetes?
Author Response
Thank you to the reviewer for their insightful comments, which have genuinely improved this manuscript.
To better organize the manuscript, we have now included the original headings of Extracellular Matrix Production, Conversion to a Myofibroblast Phenotype, and Proliferation and Migration as sub-headings under the main heading of Cardiac Fibroblast Phenotype. This has changed the numbering of subsequent sections.
The authors should discuss whether the diabetic fibroblast phenotype is limited to one specific phenotype for all stages and degrees/types of diabetes? Cardiac fibroblasts display a range of phenotypes in other cardiac diseases, depending on the stage of disease and various mechanical cues (e.g. Fu et al., JCI 2018, Herum et al., JCM 2017). The stage of disease could potentially also explain conflicting results in findings regarding the myofibroblast phenotype in cardiac fibroblasts isolated from diabetic hearts (section 3 page 2).
This is an important point. We have now incorporated an extensive discussion into section 2.3 dealing with the possibility that cardiac fibroblast phenotype changes depending on the stage of remodeling in the diabetic heart. We have incorporated the articles suggested by the reviewer, however, to our knowledge there are no studies that specifically investigate this in the setting of diabetes.
The authors should also discuss the limitations of in vitro studies of cardiac fibroblasts. Primary cardiac fibroblasts change phenotype during culturing on plastic. Thus, there may be phenotypic differences in the various in vitro studies depending on culturing time. Are there any studies investigating HG effects on cardiac fibroblasts on hydrogels with in vivo-like stiffness?
We have now incorporated discussion of the limitations of in vitro studies into section 2.3 Proliferation and Migration, and section 5 Limitations of the Literature and Future Directions. We have only been able to identify one study that used collagen I as the substrate for cardiac fibroblast culture under high glucose conditions (ref 22). Other than that we have not been able to find any studies using hydrogels and high glucose.
Please summarize the part about proliferation on page 3, line 117. In the conclusion the authors state that HG increases proliferation. This was not clear when reading the part about proliferation.
In the final paragraph of the Proliferation and migration section we have added proliferation as one of the phenotype changes that the data are not clear about. We have also changed to conclusion to indicate that it is not clear as to whether cardiac fibroblasts proliferate in response to high glucose.
Page 4, line 156. Tranilast can inhibit several other pathways than TGFb. Has it been verified to specifically inihibit TGFb? If so the reference should be given. Tranilast can inhibit effects caused by TGFb, but the mechanism is not known.
We have edited the sentence to reflect inhibition of the production of TGF by tranilast rather than inhibition of TGF itself. “Incubation with the inhibitor of TGF-bproduction, tranilast, reduced [3H]-proline incorporation by cardiac fibroblasts under HG conditions, indicative of a role for TGF-b1 in fibroblast proliferation [42].”
If possible, section 5.2 and 5.3 would be improved by including a final sentence summarizing each section.
In re-formatting the manuscript section 5.2 is now section 3.2 and section 5.3 is now section 3.3. Section 3.2 now has the following summary sentence, “…increased active TGF-bappears to be a consistent response to HG, resulting in increased ECM production, proliferation, and myofibroblast transition.” Section 3.3 now has the following summary sentence, "Thus, HG conditions cause activation of a number of kinases potentially including ERK1/2, p38, AMPK, and Akt. These kinases appear to be important in promoting a pro-fibrotic fibroblast phenotype in response to HG, but may show species-specific differences.”
The authors should discuss and include the paper by Romaine et al., describing fibrosis in integrin alpha 11 overexpressing mice. (Romaine et al., Acta Physiol, 2018)
The following text has been added to the section of the manuscript discussing the effect of high glucose on integrins, “Interestingly, overexpression of a11integrin (90-fold) in non-diabetic mice causes cardiac fibrosis when examined in mice 6 and 12 months of age, with some overlap of areas with a11integrin and collagen deposition [51]. TGF-b2 and TGF-b3, but not TGF-b1, mRNA were up-regulated in these a11integrin overexpressing mice.”
Page 5, line 217. The authors could discuss other studies finding opposite regulation of collagen I and III, which has also been observed in vivo (Sullivan et al., Stem Cell Res 2014) and in vitro (Herum et al., MBoC 2017) in response to matrix stiffening.
We have added discussion of the recommended articles as follows, “Interestingly, mechanical stiffening of gels (from 8 kPa to 30 kPa) on which isolated cardiac fibroblasts were cultured under normal glucose conditions caused up-regulation of col1a1 and col2a2 mRNA, while col3a1 mRNA was reduced [40]. Similarly, in the stiffening heart post myocardial infarction, there is a shift towards increased collagen I (57% vs 16% in control) and a loss of collagen III (0% vs 2% in control) [52]. Thus, mechanical ECM changes may also contribute to collagen up-regulation, including in diabetes.”
Several places in the text, it is stated that “this was not due to osmotic effects of glucose”. Does this mean that in many studies, the osmotic effects of glucose are not controlled?
The addition of glucose to cell culture media increases osmolarity of the media. This in itself could be a stimulus to induce changes in fibroblast phenotype rather than a specific effect of glucose on the cell. Several studies tested the effect of the increased osmolarity by also treating fibroblasts with mannose. These studies show that it is not increased osmolarity that induces changes in fibroblast phenotype, but rather that it is the increase in glucose specifically that is responsible. Many studies also use dextrose in place of glucose because it does not increase osmolarity.
Sections 5.9 – 5.11 should be explained more clearly to help the reader understand the important messages. E.g. line 336-338, line 351-352 (“the same was true for … , indicating…).
Under the revised formatting of the manuscript these are now sections 3.9 – 3.11. We have added new text to attempt to better explain the meaning of the findings reported in these sections.
Although it is good that the limitations of isolated cardiac fibroblast experiments is mentioned, the last part of the conclusion is not very summarizing of the review. The authors could add a separate section mentioning these limitations since most of the reviewed literature is based on in vitro cardiac fibroblast experiments. This sections could also include the aforementioned limitations of cardiac fibroblast spontaneous activation and differentiation in vitro.
We have added a new section to the manuscript (section 5) titled, “Limitations of the Literature and Future Directions”. This means that the final Conclusion section is now more of a summary of the literature.
Since it is not known whether high glucose induces a myofibroblast phenotype, Figure 1 and Figure 2 should read "cardiac fibroblast proliferation" instead of "myofibroblast proliferation". Figure 2 could be compacted to reduce empty space.
The suggested revisions have been made to the Figures.
Minor comments:
Page 2, line 45 "pro-phenotype" should be "pro-fibrotic phenotype"Correction made
Page 2, line55. Was collagen I production increased in diabetic- compared to non-diabetic patients? Please clarify.
Yes, the increase in collagen I is in fibroblasts from diabetic individuals compared to non-diabetic fibroblasts. This has been clarified in the text.
Page 3, line 95. Should it read “This difference was greater”?
Apologies for the poorly written description. The sentence has now been changed to, “Similarly, cardiac fibroblasts isolated from Zucker obese diabetic rats show an increased proliferative rate compared to the lean controls when grown on a collagen I matrix [27].”
Page 3, line 110. Please specify whether the patients studies in ref 36 had left ventricular dysfunction
Ref 36 (now ref 37) were individuals who underwent cardiac surgery for coronary artery bypass or valve replacement. None had diabetes mellitus, although all had related conditions such as hypertension and dyslipidemia. The cardiac fibroblasts were isolated from these individuals and then treated with high glucose. Unfortunately, no information is provided as to left ventricular function. In going back and looking closely at this study and the other study using human atrial fibroblasts that we originally compared it with, there are other factors that may account for the differences in response. We have amended the sentence in the text to read, “Some possible reasons for the differences in findings between these two studies include: 1) differences in gender composition of the individuals from which the fibroblasts were derived (4 females and 3 males in [37]versus 1 female and 11 males in [26]); 2) the atrial fibroblasts were from non-diabetic individuals in [37]and subsequently treated with HG media, whereas the atrial fibroblasts were derived from diabetic individuals and cultured in normal media in [26]; and 3) proliferation was assessed at 1 day in [37], whereas the first time-point assessed in [26]was at 2 days (final time-point = 7 days). Thus, it is possible that there may be an acute proliferative response (1 day) that then returns to normal by 2 days onwards.”
Page 3 line 122. “This was despite up-regulated beta1 integrin…”: Increased adhesion will often lead to less migration.
The sentence has been removed.
Page 4 line165. Please state what losartan is for non-clinical readers.
The sentence now reads, “…the AT1receptor antagonist losartan…”.
Page 4 line 180. Define abbreviation AMPK. Also in list of abbreviations.
Changes made.
Page 6 line 281. Sentence starting “In the STZ model…” is confusing. Please rephrase.
The sentence has been changed to, “In the STZ model of diabetes, long-term administration of exogenous IL-33 (5 weeks) attenuated collagen type I deposition as well as improved cardiac function [54].”
Page 6 Add reference for line 283-289
Reference added.
Page 7 line 292. Are these adult rat cardiac fibroblasts compared to the neonatal rat fibroblasts mentioned in line 295? Please state adult/neonatal in line 292.
Yes. The general concept was that general MMP activity was downregulated in rat cardiac fibroblasts by high glucose, however, because MMP-2 and MMP-9 were up-regulated in rat and mouse cardiac fibroblasts, then it may be that specific MMPs are upregulated (i.e. MMP-2 and MMP-9). We have attempted to clarify this in the manuscript. We have now specified that the rat cardiac fibroblasts are adult cells.
Relaxin has been shown to act through AT2 receptors. Are there any studies showing anti-fibrotic effect of AT2 activation in diabetes?
Female diabetic Zucker rats have been reported to have a reduction in AT2 receptor mRNA levels, which does not occur in males (Lum-Naihe et al., Sci Rep, 2017;7:17823). Although lean Zucker males have a lower expression level to begin with. To our knowledge there are no studies showing anti-fibrotic effects of AT2 receptor activation in the diabetic heart.
Reviewer 2 Report
This manuscript described the underlying mechanism of cardiac fibroblast during the progression of diabetes. Overall, the review is well written, organized, and included updated literature. The information provided by the author may provide a platform for future research in diabetes and/or diabetic complications. This paper can be published in its current form with minor revision. It maybe more informative to include a paragraph discussing clinical observations and the challenges for clinical translations, as most of the studies described in this review involved largely in vitro or animal studies.
Author Response
Thank you to the reviewer for their suggestion, which has genuinely improved this manuscript.
To better organize the manuscript, we have now included the original headings of Extracellular Matrix Production, Conversion to a Myofibroblast Phenotype, and Proliferation and Migration as sub-headings under the main heading of Cardiac Fibroblast Phenotype. This has changed the numbering of subsequent sections.
It maybe more informative to include a paragraph discussing clinical observations and the challenges for clinical translations, as most of the studies described in this review involved largely in vitro or animal studies.
Thank you for the suggestion. We have added a new section to the manuscript (section 5) titled, “Limitations of the Literature and Future Directions” that discusses the challenge of translating findings from in vitro studies to the clinical arena.